# Going Vegan for the Gain: A Cross-Sectional Study of Vegan Diets in Bodybuilders during Different Preparation Phases

**DOI:** 10.3390/ijerph20065187

**Published:** 2023-03-15

**Authors:** Stefano Amatori, Chiara Callarelli, Erica Gobbi, Alexander Bertuccioli, Sabrina Donati Zeppa, Davide Sisti, Marco B. L. Rocchi, Fabrizio Perroni

**Affiliations:** Department of Biomolecular Sciences, University of Urbino Carlo Bo, 61029 Urbino, Italy; stefano.amatori1@uniurb.it (S.A.); c.callarelli@campus.uniurb.it (C.C.); erica.gobbi@uniurb.it (E.G.); alexander.bertuccioli@uniurb.it (A.B.); sabrina.zeppa@uniurb.it (S.D.Z.);

**Keywords:** bodybuilding, muscle growth, plant-based diet, protein, strength, vegetarian

## Abstract

Numerous athletes compete at a high level without consuming animal products; although a well-planned vegan diet might be appropriate for all stages of the life cycle, a few elements need to be addressed to build a balanced plant-based diet for an athlete, particularly in bodybuilding, in which muscle growth should be maximised, as athletes are judged on their aesthetics. In this observational study, nutritional intakes were compared in a cohort of natural omnivorous and vegan bodybuilders, during two different phases of preparation. To this end, 18 male and female bodybuilders (8 vegans and 10 omnivores) completed a food diary for 5 days during the bulking and cutting phases of their preparation. A mixed-model analysis was used to compare macro- and micronutrient intakes between the groups in the two phases. Both vegans and omnivores behaved similarly regarding energy, carbohydrate, and fat intakes, but vegans decreased their protein intake during the cutting phase. Our results suggest that vegan bodybuilders may find difficulties in reaching protein needs while undergoing a caloric deficit, and they might benefit from nutritional professionals’ assistance to bridge the gap between the assumed proteins and those needed to maintain muscle mass through better nutrition and supplementation planning.

## 1. Introduction

Nowadays, numerous athletes compete at a high level without consuming animal products, guided by ethical and health reasons [1,2]. A well-planned vegan diet that considers the management of critical nutrients could be appropriate for all stages of the life cycle, as well as for athletes [3], as there seem to be no differences between plant-based and omnivorous diets when comparing physical performance [4].

However, a few elements need to be addressed to build a balanced plant-based diet for an athlete. Plant-based diets are known to promote early satiation and reduced appetite due to many plant foods’ high amounts of fibre and low-calorie density. While this could be positive while trying to lose weight, it can become an issue when the athlete’s energy balance needs to be managed [5]. Vegan athletes should pay attention to the quantity and quality of the protein ingested, given that plant-based proteins are often incomplete: Indeed, plant-based proteins typically contain fewer essential amino acids (EAAs) than their animal-based equivalents [6]. Nonetheless, combining different plant protein sources makes it possible to improve the overall quality of protein meals since the amino acid composition of the various plant proteins can complement each other [3]. In addition, the digestibility of plant-based proteins appears to be significantly lower than that of animal products due to anti-nutritional factors such as phytic acid and trypsin inhibitors, oxalates, phenolic compounds, and tannins, which limit the absorption of nutrients [7,8]. However, simple domestic preparation methods such as cooking, germination, fermentation, soaking, and dehulling can reduce the anti-nutritional factors improving protein bioavailability [9].

A properly designed vegan diet seems to be able to meet the protein needs of an endurance athlete via whole foods alone; however, this may not be ideal for maximising muscle growth, as the protein demands for these objectives are higher [5,7]. Indeed, plant protein quality has been reported to likely limit the stimulation of muscle protein synthesis and, subsequently, the gains in muscle mass [10,11]. In competitive bodybuilding, athletes are judged on their aesthetics: muscle size, muscular proportions, and conditioning (the absence of body fat). To achieve the required physique, athletes undertake intensive resistance training and dietary manipulations to increase muscle mass and reduce fat mass. The nutritional strategies vary according to an athlete’s competitive cycle phase. The two primary phases of a bodybuilder’s competitive cycle are the muscle-gaining phase, also called “bulking”, and the contest preparation phase, also known as “cutting”. During the bulking phase, the goal is to increase muscle mass without adding unnecessary body fat. To achieve that, athletes resort to resistance training and the maintenance of a positive energy balance, with a protein requirement ranging from 1.6 to 2.2 g/kg of body mass [12]. The cutting phase involves a reduction in body fat and the maintenance of the muscle mass gained through the bulking phase. During this phase, in addition to regular resistance training, most bodybuilders follow a high protein (2.3–3.1 g/kg of body mass), calorie-restricted diet, aerobic exercise, and isometric “posing practice” to prepare for the mandatory physique poses that judges use to place competitors [13,14].

To reach the high protein targets, “traditional” bodybuilding diets include a considerable amount of animal-source food and resort to high-quality protein supplements such as whey protein powder [15]. Whey proteins are considered to be high-quality proteins due to their digestibility and quantity of essential amino acids, providing the proteins in correct ratios for human consumption [16]. This presents a challenge for bodybuilders following a vegan diet since vegans avoid common protein sources, such as meat and dairy products. Foods high in plant-based proteins such as seitan, tofu, tempeh, and new meat alternatives can be beneficial while building a high-protein diet, and supplemental vegan protein powders can also help meet the needs by providing concentrated sources of protein surrounding workouts and throughout the day [7,17]. Emerging data support the efficacy of vegan options of protein powder in fostering muscle hypertrophy post-resistance exercise, improving indices of body composition and exercise performance [18,19,20].

With these premises, we investigated whether it was possible to follow a vegan diet and, at the same time, meet the nutritional needs of bodybuilders. Most of the scientific literature concerning vegan diets in athletes is limited to endurance sports. This observational study compared nutritional intakes in a cohort of natural omnivorous and vegan bodybuilders. The study aimed to highlight the differences between vegan and omnivorous bodybuilders in the intake of macronutrients and micronutrients at different stages of preparation for a bodybuilding competition.

## 2. Materials and Methods

### 2.1. Participants

For this study, a convenient sample of competitive bodybuilders from northern Italy was invited to participate through word of mouth and social network announcements. Twenty-two subjects expressed their interest, twenty accepted to participate after receiving the procedures’ explanation, and eighteen correctly completed the data collection. All subjects had been practising bodybuilding for several years, and they were affiliated with the Italian World Natural Bodybuilding Federation (WNBF). The participants were divided according to their diet into a group of vegans and a control group of omnivores. All participants were informed of the study’s aims and signed a written informed consent form to participate. The study was approved by the Human Research Ethical Committee of Urbino University (approval no. 31_2020) and was conducted in accordance with the Declaration of Helsinki.

### 2.2. Data Sources and Measurements

The participants completed an initial online questionnaire on anthropometric measurements, years of training and competition, training hours per week, and diet type. The participants were asked to record their food intake on an online food diary over a 5-day period at two points during the competition preparation, the muscle-gaining phase (bulking), and the contest preparation phase (cutting), in order to be able to highlight the differences in dietary composition between the two phases. The period for completing the diary differed for the athletes depending on each competitive calendar; the data collection period during the bulking and cutting phases was two months apart for all athletes. Food quantities were recorded in grams and/or portions, and dietary/sports supplement consumption was also requested; the missing data from the questionnaire and clarifications on food consumed/portions were followed up via email or phone call. The dietary analysis of the participants’ diets was performed using the WinFood (WinFood, San Giovanni, Teramo, Italy) nutritional analysis software. The daily intake of macronutrients was reported in grams per kilogram of body weight (g/kg/day).

Micronutrients such as calcium, sodium, and iron were calculated in milligrams per kilogram of body weight per day (mg/kg/day), while vitamin D and vitamin B12 in micrograms per kilogram of body weight per day (mcg/kg/day). The investigated micronutrients were selected as being of critical importance for a plant-based diet and of greatest interest for testing differences between diets. The macronutrients of the supplements were included in the analysis based on the manufacturer’s specifications on the brands’ websites.

Different sources were used to compare whether the dietary intakes (calories and macro- and micronutrients) were in line with the recommended dietary allowances (RDAs) present in the literature. In the case in which specific recommendations for bodybuilders were present, such as for protein, sport-specific recommendations were chosen [12,14]. Regarding the dietary intakes of critical micronutrients, as no specific recommendations for bodybuilders were present, the RDAs provided by the Italian Society of Human Nutrition for omnivores [21] and vegetarian/vegan diets [22] were used.

### 2.3. Statistical Analyses

Data were reported as mean ± standard deviation or median (first and third quartile) when appropriate. A multivariate analysis of variance was conducted to compare the characteristics of the two groups at baseline. Comparisons between omnivorous and vegan athletes in the two preparation phases (bulking vs. cutting) were performed using a mixed-model analysis, with athletes as a random factor, diet (omnivore vs. vegan) and preparation phase (bulking vs. cutting) as fixed factors and days (1 to 5 days of dietary recording) as a repeated measure; macronutrients and micronutrients were the dependent variables. The effects of diet, phase, and diet × phase interaction were examined. Furthermore, a one-sample Hotelling T^2^ test was used to test if athletes respected the RDAs (expected values) according to their diet and the preparation phase; post hoc analyses were performed using one-sample *t*-tests with Bonferroni correction. Statistical significance was set at alpha = 0.05, and the analyses were performed using SPSS Statistical Package for Social Sciences v26 (IBM, Armonk, NY, USA) and RStudio 4.1.1 (Posit, Boston, MA, USA).

## 3. Results

### 3.1. Participants’ Characteristics

In total, 18 bodybuilders (11 males and 7 females; age = 34.8 ± 6.4 years old; height = 173.3 ± 8.9 cm; body mass = 72.3 ± 12.6 kg; fat mass percentage = 12.3 ± 3.4%) took part in this study. The participants had been practising bodybuilding for 6 years (Q1–Q3: 3.2–7) and trained for 8 h per week (Q1–Q3: 7–8). After the initial questionnaire, they were divided according to their dietary regimen into vegans (5 males and 3 females) and controls (omnivores; 6 males and 4 females). The vegans reported being on a plant-based diet for 2 to 11 years. No significant differences were reported at baseline between the two groups for none of the investigated characteristics.

### 3.2. Dietary Differences between Omnivore and Vegan Bodybuilders

The mean macronutrient, micronutrient, and energy intake scaled for body mass are reported in Table 1 and were separately examined for the bulking phase and cutting phase of the competitive cycle.

The results revealed a significant reduction in many dietary components as preparation progressed (phase effect). A reduction in calories (*F*_(1,160)_ = 956.8, *p* < 0.001), carbohydrates (*F*_(1,160)_ = 510.4, *p* < 0.001), fats (*F*_(1,160)_ = 56.6, *p* < 0.001), and its saturated, monounsaturated, and polyunsaturated components (*F*_(1,160)_ = 7.1, *p* = 0.009; *F*_(1,160)_ = 21.1, *p* < 0.001; *F*_(1,160)_ = 5.8, *p* = 0.017, respectively) was observed between the preparation phases of both groups. A significant reduction in calcium (*F*_(1,160)_ = 9.1, *p* = 0.003), iron (*F*_(1,160)_ = 16.7, *p* < 0.001), and vitamin D (*F*_(1,160)_ = 5.4, *p* = 0.022) was also observed between the phases.

Omnivores consumed significantly more protein than vegans (*F*_(1,16)_ = 5.4, *p* = 0.033). The analysis indicated greater protein intake scaled to body mass among omnivores compared with vegans, and these groups changed the protein intake differently throughout the preparation phases (diet × phase: *F*_(1,160)_ = 11.6, *p* = 0.001). Indeed, omnivores increased their protein intake in the cutting phase compared with the bulking phase (2.24 vs. 2.57 g/kg/day), while the vegans reduced it (2.23 vs. 1.78 g/kg/day). Although no difference was observed in the consumption of carbohydrates between the different diets, its reduction over preparation phases revealed significant differences between the omnivore and vegan groups (diet × phase: *F*_(1,160)_ = 11.6, *p* = 0.001). No other significant differences were detected between the two groups.

### 3.3. Achievement of Recommended Dietary Allowances

The dietary intakes were investigated on whether they were in line with the recommendations present in the literature. The thresholds provided by Iraki et al. [12] for the bulking phase and Helms et al. [14] for the cutting phase about protein intake in bodybuilders were used. Moreover, the dietary intakes of critical micronutrients were compared with the RDAs provided by the Italian Society of Human Nutrition (SINU) for omnivores [21] and vegetarian/vegan diets [22]. The Hotelling T^2^ tests were all statistically significant for all the groups tested (omnivore bulking, omnivore cutting, vegan bulking, and vegan cutting) (*p* < 0.01). The sample data are expressed with 95% confidence intervals.

#### 3.3.1. Protein

During the bulking phase, the omnivore group consumed protein ranging from 2.2 to 2.7 g/kg/day, while vegans consumed 1.9–2.5 g/kg/day. The results revealed that both omnivores and vegans during the bulking phase reached the recommended protein range of 1.6–2.2 g/kg/day, and in some cases, these values were exceeded (*t* = 3.77, *p* = 0.04 for omnivores; *t* = 1.59, *p* = 0.16 for vegans). The recommended protein ranges for the cutting phase were 2.3–3.1 g/kg/day; vegans (1.6–2.2 g/kg/day) failed to reach the protein requirements (*t* = −5.42, *p* = 0.001), while the omnivores (2.2–2.7 g/kg/day) remained within this recommendation (*t* = −1.78, *p* = 0.11). A graphic representation of the protein intake in relation to guidance is shown in Figure 1.

#### 3.3.2. Calcium

The RDA for calcium is 1000 mg/day regardless of the type of diet and phase; neither omnivores nor vegans reached this value during bulking (384–694 mg/day vs. 336–567 mg/day) or cutting (313–573 mg/day vs. 310–505 mg/day) (*p* < 0.001 for all comparisons).

#### 3.3.3. Vitamins

The RDA for vitamin D is 15 mcg/day regardless of the type of diet and phase. Our results showed that the intake of vitamin D in the omnivore and vegan groups was highly variable and shifted over a wide range, including RDA cut-off in three of four observations, which were omnivore bulking (1.1–22.3 mcg/day), vegan bulking (5.9–31.9 mcg/day), and vegan cutting (5.7–31.8 mcg/day), while omnivore cutting failed to include the cut-off (0–12.7 mcg/day; *p* < 0.05). Our observations of vitamin B12 intake were considerably different, despite none of the two groups reaching the RDA cut-off for B12 (2.4 mcg/day) (*p* < 0.001). Notably, the results showed that while omnivores had little B12 intake in both bulking and cutting (0–0.92; 0.06–0.95 mcg/day), the vegan group had largely higher intake during both phases due to the daily supplementation of 50 mcg of vitamin B12 by all the athletes.

#### 3.3.4. Iron

For omnivore adults, iron intakes of 10 mg/day for males and 18 mg/day for females are suggested [21]. Recommendations for vegetarians indicate that iron intakes increased by 80%, achieving 18 mg/day for males and 33 mg/day for females [22]. Omnivore males’ intake exceeded RDA (13.9–17.9 mg/day) (*t* = 3.35, *p* = 0.006), while omnivore females failed to reach it (9.3–11.5 mg/day) (*t* = −7.84, *p* < 0.001). Vegan males reached the recommended iron intake (18.8–25.2 mg/day) (*t* = 1.41, *p* = 0.19), and in some cases, these values were exceeded; by contrast, vegan females did not reach RDA (11.7–17.9 mg/day) (*t* = −6.93, *p* = 0.001).

#### 3.3.5. Zinc

Both male and female omnivores’ zinc intake (10.4–13.8 mg/day; 7.7–10.8 mg/day) did not differ from the one recommended (12 mg/day for males; 9 mg/day for females). Vegan males and females failed to reach zinc RDA (18 mg/day for males; 13.5 mg/day for females) with an intake of 8.6–12.7 mg/day (*t* = −3.94, *p* = 0.003) and 3.2–6.4 mg/day (*t* = −6.40, *p* = 0.001).

## 4. Discussion

This study aimed to determine if the intakes of macronutrients and micronutrients differ between omnivore and vegan bodybuilders at different stages of preparation for competition. As expected, the energy intake of all the athletes was higher at the start of the competitive cycle during the bulking phase than that at the end of the cutting phase, as reported in previous studies [23]. Both vegans and omnivores behaved similarly when looking at their energy intake. The average caloric reduction for the omnivore group was 8.5 kcal/kg/day, corresponding to a deficit of 700 kcal/day for males and 510 kcal/day for females. Likewise, the vegans, who had shown an average caloric reduction of 7.9 kcal/kg/day, reached a deficit of 665 kcal/day for males and 385 kcal/day for females. In contrast, two case studies of bodybuilders reported a higher reduction in energy intake between 882 and 1300 kcal [24,25], while others had more similar results, reporting 554 kcal [26]. Smaller reductions in caloric intake are preferred rather than higher deficits, as they are intended to counteract metabolic adaptations to dieting and better preserve muscle mass. The caloric deficit should result in a weight loss of approximately 0.5% to 1% weekly, and this dieting period should be tailored to the competitor based on the starting body fat percentage [14]. The weekly weight loss in the present study was estimated to be 0.83% in omnivore males, 0.88% in omnivore females, 0.86% in vegan males, and 0.66% in vegan females: These values were within the recommended range, meaning that both groups respected weight loss guidance for preserving lean body mass. A summary of dietary recommendations for bodybuilding in the two main phases of the competitive cycle is reported in Table 2.

An adequate carbohydrate intake during the competitive cycle is needed to maintain muscle glycogen and enhance resistance training performance [27]. As expected, carbohydrate was the most abundant macronutrient consumed across all the preparation phases in both vegans and omnivores. Their intake during the bulking phase was in line with the recommendations indicated by Iraki et al. [12] (≥3–5 g/kg/day) and was reduced from the start to the end of contest preparation, reflecting the practice previously reported in bodybuilding case studies [24]. In our cohort, omnivores had reduced carbohydrates more than vegans while transitioning into the cutting phase, shifting from 5.1 to 3.5 g/kg /day compared with 4.8 to 3.4 g/kg/day for vegans.

The protein intakes derived from whole foods and supplements were between 2.2 g/kg/day and 2.7 g/kg/day amongst bulking omnivores and between 1.9 g/kg/day and 2.5 g/kg/day amongst bulking vegans reaching, and for the majority exceeding, the recommended protein range (1.6–2.2 g/kg/day) suggested by Iraki et al. [12]; however, these values are similar to reports from other case studies (2.2 to 3.5 g/kg BW) [25,28]. Although we did not collect data to verify muscle growth, evidence suggests that while adequate amounts of protein are consumed, the dietary protein source does not affect resistance-training-induced adaptations [29]. Since the vegan group had no problem reaching the protein requirements while bulking, this will likely reflect a successful increase in muscle mass and support of muscle strength. However, regarding protein mixtures, the non-equivalence between plant-based protein isolates and whey protein has been demonstrated, highlighting that many plant products contain insufficient amounts of leucine, which is the pivotal element in the anabolic potential. Blends of various plant-based protein sources may provide protein characteristics that closely reflect the typical characteristics of animal-based protein sources and are preferable to protein isolates from a single vegetable source [30,31]. During energy restriction in the cutting phase, prioritising protein over other macronutrients is a smart practice, as high-protein diets are known to spare muscle mass during energy deficits. Indeed, a recent systematic review on bodybuilding contest preparation recommended keeping protein intakes between 2.3 and 3.1 g/kg while cutting calories [14]. In the present study, we observed that, as suggested, the omnivore group maintained their protein intake unchanged (2.2–2.7 g/kg/day) during the cutting phase. Conversely, a protein intake of 1.6–2.2 g/kg/day was observed in the vegan group, which is below the guidelines. Furthermore, considering the above-described limitations of plant proteins (e.g., bioavailability and amino acid profile), vegan athletes should aim for protein intakes toward the higher end of the recommendations [7]. It should be pointed out that a reduction in protein after switching to the cutting phase was detected in all vegan participants; this trend, which occurred during a caloric deficit, could likely put vegan bodybuilders at risk of losing the muscle mass they successfully built during the bulking phase.

Most foods are not composed exclusively of one macronutrient, i.e., only proteins, carbohydrates, or fats, but from a combination of the three. The vegan diet does not rely on commonly known protein-rich foods, such as meat, fish, eggs, and dairy products. Vegans obtain proteins from a variety of foods whose principal macronutrients may not always be proteins, for instance, legumes and cereals [32]. While looking for a caloric deficit during the cutting phase, it is common practice to reduce the consumption of cereals, as also seen being reported in other case studies [24,33]; this may have contributed to an overall reduction in protein intake in the vegan bodybuilders because cereals would have added a non-negligible amount of protein to their diet. Noting this critical issue, a smart strategy for a vegan bodybuilder looking for a caloric deficit might be to not only reduce the consumption of typical cereals (such as rice and pasta) but to replace them with their richest protein equivalents (i.e., pseudocereals). Pseudocereals, such as quinoa, amaranth, and buckwheat, even though not frequently consumed foods, have a high protein quantity (16.3, 13.5, and 13.1 g/100 g, respectively) and a higher quality score, as they have leucine as the most abundant EAA [34]. Moreover, increasing plant-protein supplementation while in a caloric deficit might be helpful since consuming a significantly greater amount of protein from whole foods would increase energy intake. To be able to help meet the high demand for protein from plant sources, a technological approach is needed to formulate “new foods”, which might help in combining the different requirements that are difficult to achieve with conventional foods [35]. Therefore, our results align with those of Hevia-Larraín et al. [29] and suggest that a vegan diet with the supplementation of isolated plant-based proteins can achieve adequate total protein intake and be suitable for a bodybuilder aiming to increase muscle mass. However, our results also show that the dietary manipulations adopted by bodybuilders to lose fat and maintain lean mass may not be successful on a vegan diet, despite supplementation, since reaching protein intakes is challenging while cutting calories.

In both groups, fat intake was the lowest amongst the three macronutrients, and like carbohydrates, it was reduced over time, shifting from bulking to cutting (0.74 to 0.62 g/kg/day for omnivores and from 0.83 to 0.69 g/kg/day for vegans). There was a tendency for the competitors in this cohort to favour low-fat diets; those values are consistent with other cross-sectional studies of bodybuilders [13,23,26] and reflect the low end of the total energy recommendations for the fat intake proposed for bodybuilding for the bulking phase (20–35%) and the cutting phase (15–30%) [12,14]. These results point out an insignificant difference between the two diets; however, regarding the quality of fats, the vegan diet was richer in polyunsaturated fats than the omnivorous one across the phases. This result was expected, as plant-based diets are known to be richer in polyunsaturated fats compared with omnivorous diets [36].

Given the monotonous nature of bodybuilders’ diets, it was of interest to investigate the adequacy of their diets in terms of some micronutrients, which have been reported as particularly critical in vegan subjects. Both omnivorous and vegan bodybuilders did not reach the RDAs for calcium with diet in any of the two stages of preparation for competition; this can be attributed to a lack of variety in the diet, as was already observed in another study on the food selection patterns of bodybuilders [15]. Regarding iron intakes, both omnivores and vegan males successfully reached the recommended intakes, despite the guidance for vegetarians being increased by 80% since non-heme iron from vegetables is less easily assimilated by the body than the heme iron found in meat. Moreover, vegan males consumed more iron than their counterparts, confirming the tendency already observed in other observational studies comparing vegan and omnivore diets [36]. All the females participating in this study failed to reach the recommended intakes, increasing their risk of iron deficiency. The iron RDAs for omnivore and vegan females are higher than for males due to the menstrual cycle. Moreover, it is worth mentioning that the low energy availability during the cutting phase may place female athletes at risk of developing amenorrhea, leading to a multi-system dysregulation known as relative energy deficiency in sport (RED-S) [37]. On the adequacy of zinc intake, neither male nor female vegans reached the guidelines, and although both the mean intake of omnivore males and females were found to not significantly differ from the one recommended, only half of them successfully met RDAs. Vegan bodybuilders seemed aware of the necessity of supplementing vitamin B12 and cared to be explicit that they were taking appropriate daily oral supplementation. Vitamin B12 deficiency, also called cobalamin, is common among vegetarians because this essential vitamin is found only in animal foods. Vitamin B12, as a cofactor, is involved in the metabolism of amino acids, nucleic acids, and fatty acids. It plays a crucial role in producing red blood cells and bone marrow formation. Cobalamin deficiency can cause anaemia through alterations in erythropoiesis and nerve transmission via the inhibition of myelin formation [38]. The omnivore counterpart did not meet B12 RDA; however, we should point out that the nutritional analysis software used to examine the diets did not have sufficient nutritional information on all foods to estimate the content of this micronutrient. The results showed that the intake of vitamin D in the omnivore and vegan groups was highly variable due to few participants taking supplements. However, assessing vitamin D status only through diet is highly limiting, as other factors, such as the amount and intensity of sun exposure, have a greater effect on vitamin D status than a diet [39].

Due to the limitations mentioned above, the findings of this study require corroboration and should be interpreted with caution. Another limitation that should be acknowledged is that participants self-reported their anthropometric data (height, body mass, and fat percentage), so alterations in these parameters due to different instruments, operators, or techniques could not be evaluated. However, this paper provides a contemporary picture of the current bodybuilding landscape in which the vegan lifestyle is starting to make its way.

## 5. Conclusions

The appearance of vegan bodybuilders is due to the need of some athletes to combine their ethical principles with the desire to achieve a competitive muscular physique, but whether a vegan diet could be suitable for an athlete competing in bodybuilding was not assessed before. Our results suggest that vegan bodybuilders may find difficulties in reaching protein needs while undergoing a caloric deficit, and they might benefit from nutritional professionals’ assistance to bridge the gap between the assumed proteins and those needed to maintain muscle mass through better nutrition and supplementation planning. However, bodybuilders’ diets are not considered health-promoting diets and—like other dietary regimens adopted by athletes—are designed to improve performance and, in this case, to increase the potential to build a competitive muscular physique. Given the lack of research into veganism in bodybuilding, our findings will likely interest bodybuilders seeking to follow a vegan diet.

Considering the bottom line, i.e., the question “Is a vegan diet suitable for bodybuilders?”, this study may be helpful in that it revealed that a vegan diet could be suitable during the bulking phase but not in the cutting phase; it remains to be investigated whether by following a nutritional intervention, athletes on a vegan diet may be able to reach protein recommendations to preserve lean mass while undergoing a calorie deficit. This research is intended to be a starting point for further investigation of this theme.

## Figures and Tables

**Figure 1 ijerph-20-05187-f001:**
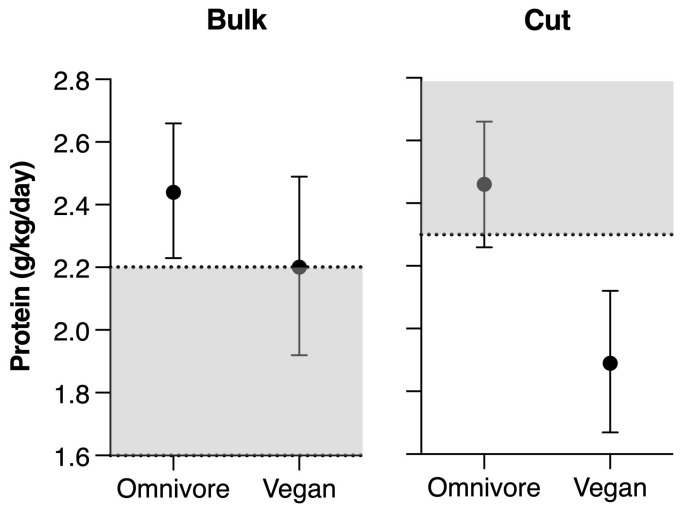
Protein intake of omnivore and vegan bodybuilders in relation to recommended values (grey area) for the two phases of the competitive cycle.

**Table 1 ijerph-20-05187-t001:** Macronutrient and micronutrient differences between omnivore and vegan bodybuilders diets. Data are reported as median (first and third quartile). *p*-values are reported for diet, phase, and interaction diet × phase effects (values in bold are statistically significant *p* < 0.05).

Variable	Omnivore		Vegan		Diet	Phase	Diet × Phase
	Bulking	Cutting	Bulking	Cutting			
Calories (kcal/kg/day)	37.23 (35.82–38.52)	28.56 (26.64–30.58)	36.41 (32.82–39.64)	27.21 (25.69–31.11)	0.383	**<0.001**	0.205
Protein (g/kg/day)	2.24 (2.18–2.65)	2.57 (2.15–2.70)	2.23 (1.87–2.45)	1.78 (1.66–2.21)	**0.033**	**0.002**	**0.001**
Carbohydrates (g/kg/day)	5.11 (4.82–6.11)	3.51 (3.01–3.96)	4.81 (3.98–5.79)	3.39 (2.93–3.84)	0.334	**<0.001**	**0.001**
Fats (g/kg/day)	0.74 (0.68–0.91)	0.62 (0.49–0.71)	0.83 (0.72–0.95)	0.69 (0.63–0.81)	0.420	**<0.001**	0.158
Saturated fats (g/kg/day)	0.06 (0.05–0.12)	0.06 (0.04–0.09)	0.08 (0.05–0.10)	0.07 (0.05–0.09)	0.403	**0.009**	0.428
Monounsaturated fats (g/kg/day)	0.25 (0.16–0.34)	0.20 (0.14–0.27)	0.24 (0.10–0.31)	0.17 (0.09–0.24)	0.494	**<0.001**	0.409
Polyunsaturated fats (g/kg/day)	0.08 (0.03–0.10)	0.05 (0.02–0.09)	0.14 (0.08–0.19)	0.14 (0.06–0.21)	**0.012**	**0.017**	0.843
Calcium (mg/kg/day)	6.71 (3.40–11.36)	6.32 (2.93–7.29)	6.49 (4.18–8.42)	5.86 (4.79–7.29)	0.659	**0.003**	0.229
Sodium (mg/kg/day)	22.00 (10.76–27.50)	19.63 (8.17–29.85)	5.96 (3.50–12.85)	5.48 (1.76–10.84)	**<0.001**	0.264	0.957
Iron (mg/kg/day)	0.19 (0.15–0.25)	0.15 (0.12–0.21)	0.28 (0.22–0.40)	0.25 (0.21–0.32)	**0.013**	**<0.001**	0.696
Zinc (mg/kg/day)	0.13 (0.10–0.18)	0.15 (0.10–0.19)	0.11 (0.05–0.18)	0.10 (0.08–0.14)	0.269	0.936	0.285
Vitamin D (mcg/kg/day)	0.00 (0.00–0.04)	0.02 (0.00–0.03)	0.00 (0.00–0.64)	0.00 (0.00–0.64)	0.262	**0.022**	**0.028**
Vitamin B12 (mcg/kg/day)	0.00 (0.00–0.00)	0.00 (0.00–0.01)	0.68 (0.63–0.80)	0.68 (0.63–0.80)	**<0.001**	0.756	0.478

**Table 2 ijerph-20-05187-t002:** Dietary recommendations for bodybuilders in the two main phases of the competitive cycle. Data extracted from Helms et al. [14] and Iraki et al. [12].

	Bulking Phase		Cutting Phase
	Novice/Intermediate	Advanced	
Weight variation (% kg/week)	+0.25–0.5%	+0.25%	–0.5–1%
Protein (g/kg/day)	1.6–2.2	1.6–2.2	2.3–3.1
Fats (% of calories)	20–35	20–35	15–30
Carbohydrates (g/kg/day)	≥3–5	≥3–5	remaining

## Data Availability

The dataset generated during the current study will be available upon request to the corresponding author.

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
