# Peer review of "Going Vegan for the Gain: A Cross-Sectional Study of Vegan Diets in Bodybuilders during Different Preparation Phases"

_ijerph, 2023, doi:10.3390/ijerph20065187_

Round 1

Reviewer 1 Report

I really enjoyed reading this work from cover to cover. I congratulate the authors for doing a study on this subject. For the environmental sustainability of our planet, it is of great importance that vegan-style nutrition is preferred more especially in athletes who consume a lot of protein. In this regard, I thank the authors again for working on this issue, congratulations !!

Author Response

Reviewer 1:

L29 Which study showed this? Just asking and wondering 1 or 2th reference?

Re: Thanks for this comment. The main reasons for people to move to a plant-based diet were mainly reported by Hopwood et al. [Ref 1], and Vitale et al. [Ref 2] reported coherent information. However, double-checking these references, we removed the word "economic" from our sentence as it would have been questionable.

Why did authors not calculate "effect sizes"?

Re: Thanks to the Reviewer for this comment. We agree that effect sizes are missing in our Results section, but SPSS does not provide estimates of the effect sizes for mixed models, as the interpretation of effect size measures in models with multiple error terms could be complicated. Effect sizes for mixed models could be theoretically calculated using hierarchical models, but we believe this procedure would overcomplicate the presentation of the results without adding any useful additional information. We hope the Reviewer will agree with us about this decision.

L142 How many females is there in groups?

Re: Thanks for this comment. The genders were divided into two groups as follows: omnivores, six males and four females; vegans, five males and three females. We have added this information in the manuscript, as it was missing.

L184-185 Speaking of menstrual cycle, it will be better to mention from "Red-S". It is your decision, just I recommend.

Re: We agree with the Reviewer that mentioning RED-S would be appropriate. Accordingly, we have added the following sentence: "Moreover, it is worth mentioning that the low energy availability during the cutting phase may place female athletes at risk of developing amenorrhea, leading to a mul-ti-system dysregulation known as Relative Energy Deficiency in Sport (RED-S) [38]." The following reference has also been added: International Olympic Committee (IOC) Consensus Statement on Relative Energy Deficiency in Sport (RED-S): 2018 Update. Unfortunately, no data about the menstrual cycle were collected in our specific female group.

Reviewer 2 Report

Thank you to give a chance for peer review to this article.  

The authors accurate describe the main stage of their research, describe the concept in details and provide all needed statistical results.

The text does not contain any major errors therefore I have some minor comments

It's not clear how you assessed the plausibility of the energy intake?

The article does not specify how many years the vegan participants have been on the diet.

Author Response

Reviewer 2:

It's not clear how you assessed the plausibility of the energy intake?

Re: Thanks for this comment. The participants were trained to fill the food diary, adding all food and drink consumed within 24 hours. Then, the diaries were checked, and any missing data was fixed by phone calling the participant. When the data were uploaded on WinFood for the analyses, the authors checked whether the results about caloric intake and nutrients were plausible; no issues were found in this phase.

The article does not specify how many years the vegan participants have been on the diet.

Re: Thanks for this comment. The vegan athletes recruited for this study followed a plant-based diet 365 days a year and reported following it for a range from 2 to 11 years. We have now added this information to the manuscript.

Reviewer 3 Report

Thank you for allowing me to review this study. I have found it to be a simple yet well-organised and straightforward study. As vegan diets gain popularity, this research area needs even more studies to be contacted in order to assist sports nutritionists and sports practitioners.

I only have a few minor comments (see below) and a few general questions.

1)      Do you have any data on the actual performance of the participants? Were one group or the other more successful in their respective competitions?

2)      The vegans followed a certain diet by choice. Did they hit their target? Did some feel that vegan diets were superior/inferior or equivalent to the omnivorous diet?

L50-51 Athletes, or exercising individuals require higher amount of protein (2 g/kg body mass) than sedentary individuals (0.8-1 g/kg body mass). This is the international consensus. Do body builders require even more protein on a daily basis? Can you add some reference for the amount of protein needed per day?

Methodology
Did the participants report their own body fat? This can vary greatly according to the method used to assess body fat. In the results section this appears quite high for body builders. How was body fat assessed, and what were the limitations in reporting the correct or actual body fat of the participants?

Results

Throughout the results section there is comparison of the findings against RDA values for the various macro- and micro-nutrients. The document is easy to follow, however, from an editorial point of view these daily cut-off points and RDA values should have been initially described in the methods sections. Could you add or amend accordingly the methods section?

In the methods section it is mentioned that the questionnaire included points on nutritional supplements intake. However, there is no data reported in the results section, and only a few words on the subject are included in the latter stages of the discussion. Do you have any data to report on use of nutritional supplements?

Discussion

L76-77 units are kcal/kg/day or just kcal/kg?

There is some discussion on nutritional supplements, but personally I would like to see more info on nutritional supplements intake.

Author Response

Reviewer 3:

Do you have any data on the actual performance of the participants? Were one group or the other more successful in their respective competitions?

Re: Thanks for this interesting question. In this study, we could not collect direct data on the physical performance of the athletes. However, following the study period, three athletes from the vegan group, more precisely two men and one woman, won respectively 1st place 2022 WNBF Dubai PRO Cup Men's Physique category, 1st place 2022 WNBF ITALY Overall Men's Physique category and 1st place in 2022 WNBF PRO Bikini category.

However, we decided not to present this information in the manuscript for two main reasons: 1) more important, this would affect the anonymity of the participants, as it would be easy to trace their identity; 2) we

feel that it would not be correct to “force” a relationship between these competition results and diet, as the excellent achievements obtained by these athletes are likely the result of several years of athletic preparation, not only of their diet.

The vegans followed a certain diet by choice. Did they hit their target? Did some feel that vegan diets were superior/inferior or equivalent to the omnivorous diet?

Re: Thanks for this question. Unfortunately, we did not collect any information about this topic.

L50-51 Athletes, or exercising individuals require higher amount of protein (2 g/kg body mass) than sedentary individuals (0.8-1 g/kg body mass). This is the international consensus. Do body builders require even more protein on a daily basis? Can you add some reference for the amount of protein needed per day?

Re: Thanks for this comment. As suggested, we have added the protein requirements of bodybuilders in the Introduction section. As can be noted in Table 2, protein intake recommendations do not differ a lot from the usual 2 g/kg for exercising individuals during the bulking phase but are higher during the cutting phase of the preparation. We reported this information according to the works of Iraki et al. [12] and Helms et al. [14].

Methodology

Did the participants report their own body fat? This can vary greatly according to the method used to assess body fat. In the results section this appears quite high for body builders. How was body fat assessed, and what were the limitations in reporting the correct or actual body fat of the participants?

Re: Thanks to the Reviewer for this comment. The body fat percentage was assessed using skinfolds (measurements were performed by each athlete's staff) and self-reported by the athletes in the questionnaire. We are aware that this technique is operator-dependent, and results can vary according to the equation used, which we don’t know about. Consequently, we have added this point among the limitations of the paper. However, the measures were taken all on the bulk phase (when the fat percentage is higher, as athletes came from the off-season), and the high fat percentage that can be observed is also influenced by the fact that data come from both males and females, with the latter notoriously having higher fat mass compared to men.

Results

Throughout the results section there is comparison of the findings against RDA values for the various macro- and micro-nutrients. The document is easy to follow, however, from an editorial point of view these daily cut-off points and RDA values should have been initially described in the methods sections. Could you add or amend accordingly the methods section?

Re: We agree with the Reviewer on this point. As requested, we have added the information about sources and procedures used for the comparisons with RDAs in the Methods section.

In the methods section it is mentioned that the questionnaire included points on nutritional supplements intake. However, there is no data reported in the results section, and only a few words on the subject are included in the latter stages of the discussion. Do you have any data to report on use of nutritional supplements?

Re: Thanks to the Reviewer for this comment. As reported in the Methods section, participants were asked to add their supplement intake to the daily food diary. However, the athletes reported no regular supplement consumption except for supplementation of vitamin B12 from the vegan athletes (in the form of cyanocobalamin for a dose of 50 mcg/day). This information is reported in the relevant section (3.3.3. Vitamins). On a few occasions, participants reported consumption of protein powders or vitamin D, but as it was not a consistent intake throughout the days, the differences in doses, frequencies and supplements sources, we opted not to dedicate a whole section to discuss this point. The absence of adequate protein supplementation in the vegan group could be one of the reasons leading to a failure in reaching the recommendations, and as reported in the Conclusions, better supplementation planning could help these athletes. Given the scarcity of supplements reported, we treated supplements as an integral part of the diet itself. A few lines to specify the functions of B12 are reported at the end of the discussion.

Discussion

L76-77 units are kcal/kg/day or just kcal/kg?

Re: Thanks for highlighting this. We changed to kcal/kg/day.

There is some discussion on nutritional supplements, but personally I would like to see more info on nutritional supplements intake.

Re: Thanks for this comment. We have already answered this point in the comment above.